# The role of vaccination and public awareness in forecasts of Mpox incidence in the United Kingdom

Samuel P. C. Brand [1,2] ✉, Massimo Cavallaro [1,2,3], Fergus Cumming[4], Charlie Turner[4], Isaac Florence [4], Paula Blomquist[4], Joe Hilton[1,2], Laura M. Guzman-Rincon [1,2], Thomas House [5], D. James Nokes [1,2] & Matt J. Keeling [1,2,3]

Beginning in May 2022, Mpox virus spread rapidly in high-income countries through close human-to-human contact primarily amongst communities of gay, bisexual and men who have sex with men (GBMSM). Behavioural change arising from increased knowledge and health warnings may have reduced the rate of transmission and modified Vaccinia-based vaccination is likely to be an effective longer-term intervention. We investigate the UK epidemic presenting 26-week projections using a stochastic discrete-population transmission model which includes GBMSM status, rate of formation of new sexual partnerships, and clique partitioning of the population. The Mpox cases peaked in mid-July; our analysis is that the decline was due to decreased transmission rate per infected individual and infection-induced immunity among GBMSM, especially those with the highest rate of new partners. Vaccination did not cause Mpox incidence to turn over, however, we predict that a rebound in cases due to behaviour reversion was prevented by high-risk group-targeted vaccination.

The global outbreak of Mpox (historically known as monkeypox) virus (MPXV) in 2022 had its origins in sporadic cases reported in Nigeria from 2017[1]. Sustained incidence of Mpox in European and North American countries from May 2022 led to a WHO declaration of a public health emergency of international concern on 23rd July 2022[2]. Cases in Europe and North America have been predominantly in gay and bisexual men who have sex with men (GBMSM), with those who have greater numbers of sexual partners being more likely to be infected[3]. Vaccines developed to target smallpox have been shown to have reasonable (short-term) efficacy against MPXV-induced disease[4–6], and many countries have offered pre-exposure prophylactic vaccination targeted at higher-risk GBMSM individuals to control infection[7]. Towards this goal the UK Health Security Agency (UKHSA) and the Joint Committee on Vaccination and

Immunisation (JCVI) have recommended the use of the Modified Vaccinia Ankara (MVA) smallpox vaccine Imvanex (called Jynneos in the USA) for Mpox at-risk groups in the UK[8].

Mpox predominantly spreads from person-to-person through prolonged physical contact with the infectious rash, scabs and/or fluids of infected individuals[9]. Traditionally, Mpox incidence has been sporadically observed in sub-Saharan Africa following zoonotic spread from wildlife reservoirs, with uncommon person-to-person transmission being associated with household cohabitation[5]. However, as mass smallpox immunisation campaigns have declined, and the resultant background immunity to Mpox has waned, there has been a substantial increase in Mpox incidence in some sub-Saharan African countries[10,11]. In the recent outbreaks in Europe and North America, the epidemiology of Mpox has been skewed towards a far higher case frequency

[1]The Zeeman Institute for Systems Biology Infectious Disease Epidemiology Research (SBIDER), Coventry, UK. [2]School of Life Sciences, University of Warwick, Coventry, UK. [3]Mathematics Institute, University of Warwick, Coventry, UK. [4]United Kingdom Health Security Agency, London, UK. [5]Department of Mathematics, University of Manchester, Manchester, UK. ✉e-mail: sam055@mac.com

among GBMSM compared to non-GBMSM groups. This is in contrast to the traditional epidemiology of Mpox described in sub-Saharan Africa, and it is likely that Mpox has found a niche in high income countries (HICs) among individuals with high frequencies of close physical contact. Early epidemiological modelling of the transmission potential of Mpox in HICs has identified the potential for Mpox to spread among sexual contact networks due to a comparatively small number of individuals with highly frequent sexual encounters, the distribution of the number of encounters is best described by a heavy-tailed distribution[3,12]. This early concern has been confirmed, while initial modelling of Mpox without a focus on sexual contact networks has been proven over-optimistic in terms of total case load[13]. Counterbalancing the transmission risk associated with contact frequency, there is evidence that awareness of Mpox symptoms with intention to reduce risk of either spreading or contracting Mpox became widespread in various high-income countries by the second half of 2022[14–17]. In particular, at least one study has found that awareness of Mpox and intention to reduce transmission risk was higher among GBMSM people compared to the general population[17].

In this simulation study, we use a bespoke Mpox transmission model calibrated to the social structure and demography of the United Kingdom, as well as the epidemiology of Mpox infections, to make projections of future incidence over a medium-term time horizon (26 weeks ahead). The features of the transmission model include both transmission rates that account for population immunity due to naturally acquired infection and/or vaccination as well the possibility of time-varying behaviour change reducing the risk of secondary infections. We use a Bayesian method to make model parameter inference from the UKHSA linelist of identified Mpox cases in the UK between May and September 2022 (see "Methods" and Supporting Information) as well as validating the model's sequential accuracy against redacted case data (see Supporting Information). The proportion of reported MPXV cases among GBMSM is important in this modelling approach; where this data was missing from the linelist we used a gradient boosted decision tree (GBDT) trained using the metadata of MPXV cases with known GBMSM status to assign a GBMSM status probability (see Supporting Information).

The MPXV transmission model is capable of exploring a range of scenarios, although here we focus on the potential impacts of vaccinating the most at-risk GBMSM and the reduction in transmission due to self-imposed reductions in physical contacts whilst having symptoms. This approach allows us to address a number of scientifically interesting challenges with clear public health outcomes. First, to quantify the likely transmission potential of MPXV among GBMSM and non-GBMSM groups in the United Kingdom. Second, to estimate the changes in effective transmissibility over time and the associated impact on the infection dynamics. Third, to assess the benefit of the vaccination campaign in reducing cases of MPXV. Finally, to make medium-term projections of MPXV case rates given that behaviour will likely return to a pre-outbreak baseline over the coming months.

## Results

We ran Bayesian inference for the parameters and case incidence trajectories in our MPXV transmission model (see "Methods" and Supporting Information) and used the posterior parameter draws to generate model-based trajectories of cases to make counterfactual projections and to forecast likely future case trends over a medium-term period (26 weeks ahead).

### Transmission potential of Mpox in the United Kingdom

We infer that the high ratio of GBMSM cases compared to non-GBMSM cases observed in the UK is due to MPXV having a low transmission potential outside high-frequency sexual contact groups. The basic reproduction number ($R_0$) for other (non-sexual)

transmission pathways was estimated to be 0.0398 (0.00898–0.0716; 95% CI) hence we expect infection to decline in the absence of spill-over from the high sexual activity GBMSM group. In contrast, on 1st May, before any behavioural change, the basic reproduction number due to sexual contact formation in the GBMSM population was estimated at 5.16 (2.96–9.24; 95% CI, see Supporting Information S.1 for reproductive ratio and predicted case distribution details), which is composed of a 43.4% (24.9–77.7%; 95% CI) transmission risk per sexual contact, a power-law distribution for the number of new partners over time (see "Methods"), and an effective average infectious period of 6.01 days (see "Methods") (Table S.3 and Fig. S2 detail inferred values for all parameters in Supporting Information). These parameters suggest that the average number of secondary cases will be greater than one for any individuals who typically have two or more sexual contacts per week. We note that our inferred effective infectious period is shorter than typically reported from clinical observation (2–4 weeks[9]), which we interpret as a shortening of the generation time of MPXV due to transmission being less likely to occur once severe symptoms manifest. The overall reproduction number for all MPXV transmission pathways on 1st May was estimated as 5.16 (2.96–9.24; 95% CI); extremely close to the reproductive number due to only sexual contacts in the GBMSM population.

To capture elements of spatial or social structure we use a metapopulation framework to partition our population into multiple cliques with weak transmission between them. We infer the structure of this metapopulation in terms of clique sizes and find that several large subpopulations are preferred, with the largest accounting for more than 10% of the population and the largest ten subpopulations containing the majority of individuals; as such there is limited inferred population structure with the majority of the GBMSM population being well connected (see "Population structure" in "Methods", Supporting Information and Fig. S3).

### Behaviour change and reduced transmission risk of Mpox in the United Kingdom

We find significant evidence that the Mpox transmission risk associated with new sexual contacts decreased over June and July 2022. Immediately before the WHO declaration of a public health emergency of international concern on 23rd July 2022, we estimate that the reproductive number for both sexual transmission within the GBMSM community and all other transmission pathways had decreased by ~40% (Fig. 1): GBMSM $R_0 = 3.02$ (1.94–4.31; 95% CI)). Immediately after the WHO declaration our posterior mean prediction was for a further decrease in these reproductive numbers: GBMSM $R_0 = 1.98$ (1.06–3.42; 95% CI)).

We interpret the decreased transmission potential of MPXV over this period as being due to greater public awareness of Mpox disease, such that some higher-risk physical contacts that would normally have occurred were avoided in May-July 2022 by individuals with MPXV symptoms (from a modelling perspective, this can be captured by maintaining the network structure of sexual contacts but reducing the transmission risk to account for some of these contacts no-longer occurring). The level of transmission within the high-risk GBMSM groups has been further limited by immunisation, which we model as providing 70–85% protection against infection[18]. In the UK, the first person believed to be at high risk of MPXV exposure volunteered for a dose of the Imvanex/Jynneos vaccine on 16th/17th July, with estimates suggesting 5000 doses per week were offered in London throughout August[18]. This is slightly later than the observed peak in MPXV cases in the United Kingdom (11th July 2022); therefore, the evidence points towards diminishing risk of transmission due to a combination of behavioural change and population immunity due to exposure as the cause of declining MPXV cases rather than the vaccination programme.

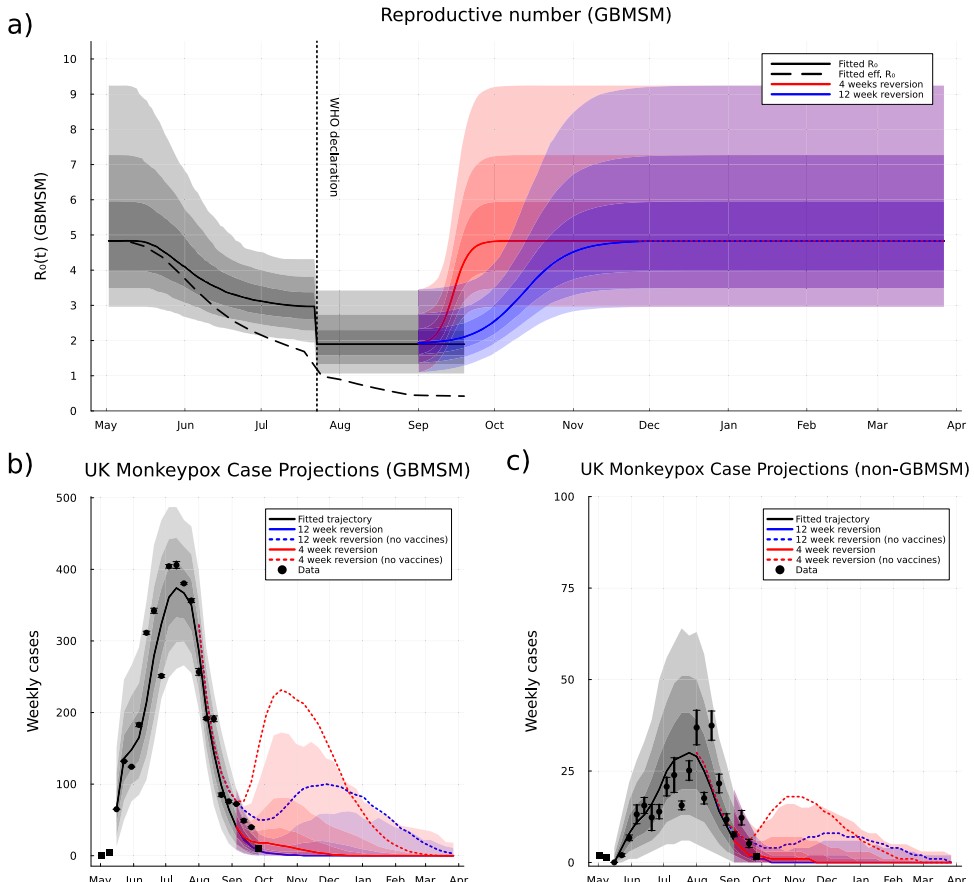

**Fig. 1 | GBMSM reproductive number and model-based case projections.** Top row: Inferred basic reproductive numbers for transmission via sexual partnership among GBMSM people (**a**). The fitted reproductive numbers are shown as basic (solid black curve), not accounting for population immunity, and effective (dashed black curve), accounting for population immunity. Reversion towards baseline behaviour is shown by basic reproductive numbers, this begins on 1st September and reaches 99% of pre-outbreak behaviour either 4 weeks (red curve) or 12 weeks (blue curve). Bottom row: Weekly Mpox confirmed case data from UKHSA linelist with inferred GBMSM status (black markers; GBMSM **b**, non-GBMSM **c**). The last

week of available data (black squares) was not used in inference. Error bars on data points indicate 95% confidence intervals for GBMSM status inference. The posterior median over model projections are shown for 4 weeks reversion and 12 weeks reversion to baseline behaviour (solid curves). The posterior median model projections for the counterfactual scenario where vaccines are either absent or ineffective generate substantial secondary peaks (dashed curves). Background shading indicates 50, 80 and 95% credible or prediction intervals; prediction intervals for the counterfactual results are omitted for clarity. Model predictions are based on $n = 2000$ parameter sets drawn from their posterior distribution.

## Nowcasting Mpox exposure and medium-term projections under behaviour reversion and continued Mpox vaccination

Sequential forecasts made during the UK outbreak, show that for early projections (formulated by fitting the Mpox transmission model to data available in late May or mid June) the median forecasts of weekly case incidence were already fairly accurate. Among the GBMSM population/group, early forecasts using data available in either late-May 2022 or mid-June 2022 captured the peak height and timing; the actual peak being 407 cases during the week starting 11th July whereas the median forecasts using early data predicted ~550 cases with a peak in the first week in July (Fig. S5). In the non-GBMSM group, early forecasts predicted a peak of ~50 cases in the week starting 11th July 2022; the observed number of cases in that week was inferred to be 24, although the true peak of 37 cases occurred some weeks later (Fig. S5). However, early forecasts had large prediction uncertainty (Fig. S5). After the observed peak in case incidence our sequential forecasts were able to accurately capture the further decline in weekly case incidence with far lower prediction variance (Fig. S5). See section S.6.2 in Supporting Information for details on sequential forecasting with redacted data.

Using all the case data for parameter inference, the Mpox transmission model was able to retrospectively capture the peak size and timing for both GBMSM and non-GBMSM, as well as the decline in weekly case incidence throughout August and September 2022.

By October 2022, our model assigns a high probability (>90%) to at least 10% of those GBMSM individuals with a high number of new monthly partners (>20 per month) having been infected with Mpox. The proportion of GBMSM who have been infected with MPXV increases with their rate of new partnerships, such that we project that the majority of GBMSM with a very high numbers of monthly partners (>39 per month) will already have been infected (Fig. 2). However, such high-risk individuals only constitute a small proportion of the population, such that by 1st October 2022 we expect only 1.12% (0.539–2.85%; 95% PI) of GBMSM and just 0.00106% (0.000211–0.00364%; 95% PI) of non-GBMSM to have been infected— further illustrating the highly skewed nature of MPX transmission in the UK. We estimate that the combined effect of natural infections and vaccinations reduced the effective reproductive number by 74.7% (59.6–94.1%; 95% PI) by the first week in September despite the comparatively few people infected with Mpox due to the centrality of high-risk individuals to the ongoing spread of Mpox.

Projecting the dynamics forward 26 weeks, from October 2022 to late-March 2023, requires making assumptions about the long-term behaviour of the population. At time of writing, we expect an eventual return to pre-outbreak mixing and behaviour, as awareness and attention declines; and we consider this reversion over two different time scales (blue and red lines in Fig. 1). In both scenarios the median forecast is for weekly case incidence to continue declining towards

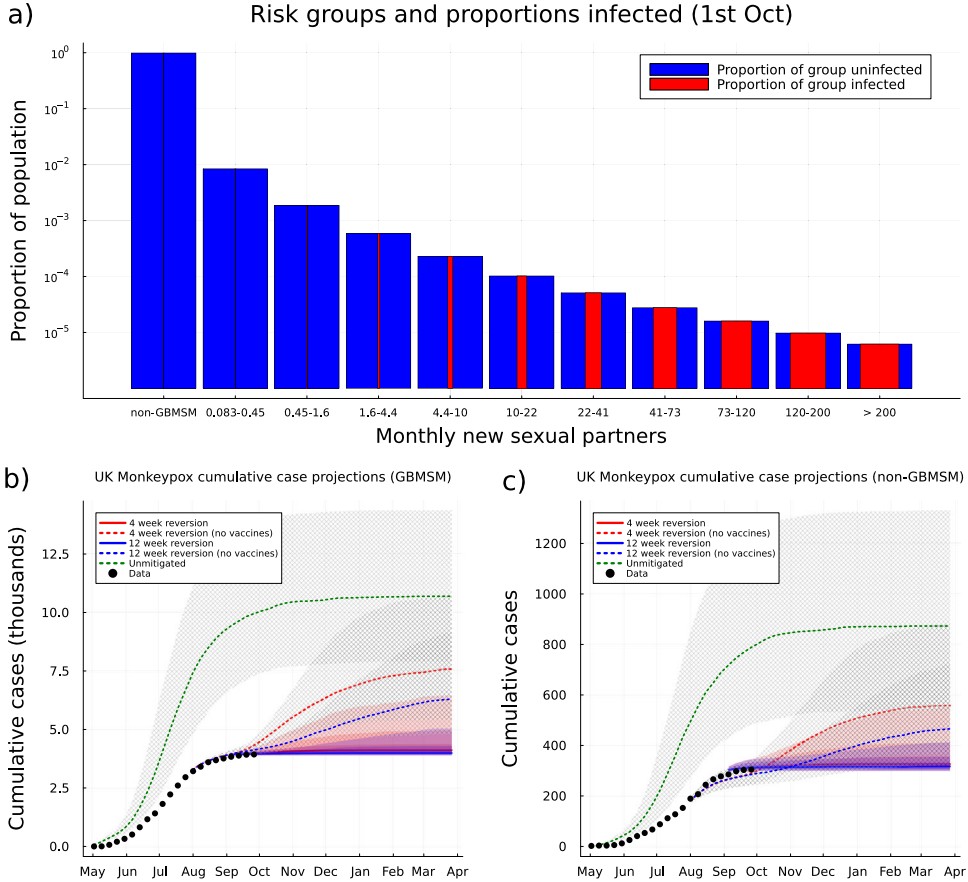

**Fig. 2 | Population exposure and medium-term cumulative case projections.**
**a** The proportion of the whole UK population in each risk group: non-GBMSM and ten GBMSM sexual activity groups (bars). The posterior median of proportion in each risk group uninfected (blue) and infected (red) by 1st October 2022 is shown as proportion shaded in each bar. Bottom row: Posterior median model projections of cumulative cases among GBMSM (**b**) and non-GBMSM (**c**) for 4-week (red) or 12-week (blue) reversion to baseline behaviour with background shading for 50, 80

and 95% prediction intervals. Also shown are counterfactual results without vaccination but with behaviour change (dashed red and blue), and without either vaccination or behavioural change (dashed green). Shaded and cross-hashed regions show the 50% prediction intervals for the counterfactual projections. Model predictions are based on $n = 2000$ parameter sets drawn from their posterior distribution.

zero by the end of 2022. However, a more prompt reversion (red) risked a rise in cases in the GBMSM population; that is the 20% and 5% of worst forecasts show a significant resurgence in cases (Fig. 1). A slower reversion (blue) makes a resurgence in cases less likely (Fig. 1). By late-March 2023, if reversion to normal behaviour occurs over 12 weeks (blue curves), we project a cumulative number of 3998 (3898–4934 95%PI) Mpox cases among GBMSM and 317 (300–412 95% PI) Mpox cases among non-GBMSM. If reversion to normal behaviour happens faster (over 4 weeks, red curves) then our projection of cumulative Mpox cases is slightly higher: 4114 (3905–6752 95%PI) Mpox cases among GBMSM and 327 (301–574 95%PI) Mpox cases among non-GBMSM (Fig. 2).

As a counterfactual scenario, we also investigated the dynamics if vaccination had not been deployed (dashed lines in Fig 1). This highlights how the decline in cases is primarily attributable to the change in behaviour and population immunity due to exposure, with the no-vaccination projections only significantly departing from the observations in September 2022. However, without the protection offered by vaccination the reversion in behaviour leads to substantial second waves of infection and cases, especially compared to the first wave in the non-GBMSM population (Fig. 1, dashed lines). By the end of March 2023 this no-vaccine counterfactual generates far more cases: reaching 7201 (4748–20,069 95%PI) for 12 week reversion and 8523 (4921–21,362 95%PI) for 4 week reversion in the GBMSM groups, and 609 (348–2005 95%PI) for 12 week reversion and 699 (360–2194 95%

PI) among non-GBMSM (Fig. 2). Given we have assumed that the vaccine has been offered to those GBMSM most at risk, which we define as those having typically more than one new sexual partner per month, the solid lines represent a lower bound on future waves while the dashed lines represent upper bounds when the vaccine is taken up by lower risk groups and therefore is less effective. Because we estimate that there were only enough doses of Imvanex to inoculate ~6.5% of the sexually active GBMSM individuals in the United Kingdom (see section "Vaccination modelling" in "Methods") randomly offering the available vaccine to any sexually active GBMSM individuals, rather than those most at risk, was nearly as ineffective as having no vaccines (see section S.6.3 in Supporting Information).

As a final counterfactual, we considered an uncontrolled outbreak without vaccination or behaviour change; this is predicted to generate 11,601 (5005–27,251 95%PI) cases by end of March 2023 (of which around 90% are in the GBMSM population) nearly three times higher than has been observed to date, with most cases observed by the beginning of October 2023 (Fig. 2).

## Discussion
We have developed a novel, stochastic, discrete population model to enhance understanding and make projections of Mpox incidence in the UK. The epidemiology of Mpox in the UK suggests an individual-based network modelling approach should also be applicable, but this more computationally intensive methodology would mean it is

challenging to make rapid inference against the continually evolving data. Our relatively simple model aimed to capture much of the essential features of a transmission network, in particular differential behaviour, whilst being feasible to fit and re-fit to available data streams, hence providing a rapid projection tool generating results of immediate benefit to policy advisors. We allow our population to be sub-divided, using a metapopulation framework to capture the partitioning of network into different cliques[19]. This metapopulation partitioning is inferred from the observed case dynamics (see "Methods"), and is sufficiently flexible to account for additional spatial or social structure; although our inference favours a well connected social structure for GBMSM partnerships.

There have been a number of recent studies finding at least the intention to change behaviour to avoid risk of Mpox transmission[14–17], with one study reporting a higher rate of intention among GBMSM people compared to the background population[17]. Our findings are consistent with such studies finding evidence of behaviour change aimed at decreasing transmission risk among GBMSM individuals. This leads us to believe that the rate of Mpox case incidence, which peaked and subsequently turned-over in mid-July 2022, was limited by behaviour change decreasing the transmission potential from GBMSM people infected with MPXV. This is supported by two factors: (1) very few vaccine doses had been deployed by mid-July and (2) we infer insufficient population immunity from natural infection to curtail the Mpox outbreak in less than 3 months without a decrease in the fundamental reproductive number. The model-based inference on the epidemic trajectory suggests that transmission potential per infected GBMSM person decreased significantly (by ~40–50%) since start of the outbreak, which we interpret as the effect of public awareness of the threat of Mpox and its symptomatic identifiability.

Our model-based analysis suggests that the most likely reason the Mpox epidemic in the United Kingdom turned over was a combination of high population exposure among the small number of people in the most sexually active groups and behaviour change resulting in relatively lower risk of forward transmission from infected people. We predict that population immunity due to a mixture of vaccination and naturally acquired infection will limit the size of the epidemic going forwards. However, we expect that behaviour changes that decreased the transmission potential of Mpox will revert towards a pre-outbreak baseline over the coming months, although predicting the speed of such behaviour change is difficult[20]. If this behavioural reversion is rapid (e.g. over 4 weeks) then it was possible that there would be a moderate resurgence in Mpox cases in the United Kingdom, however, we expect that any resurgent wave would be much smaller than the wave that peaked in mid-July.

Our expectation that the transmission potential of people infected with Mpox will increase over the coming months as behaviour reverts towards a pre-outbreak baseline underlines the likely importance of the Imvanex vaccination campaign aimed at GBMSM people deemed at higher risk of Mpox and health care professionals. Our model-based analysis suggests that the vaccine doses given whilst transmission potential was comparatively low in August and September will contribute significantly to the population immunity of the most at-risk GBMSM people, and therefore, avoids a substantial second wave of MPXV incidence in the United Kingdom. In the scenario used in this modelling study (that is where Imvanex doses have been taken up efficiently by those GBMSM people having typically more than one new sexual partner a month) we projected that the vaccination campaign will roughly halve the number of cases up until the end of March 2023 compared to alternative of no vaccines or inefficiently targeted vaccines. In particular, we find that failing to target vaccination towards the most at-risk groups would have been almost as ineffective as a no vaccination counter-factual.

Although we favour the explanatory role of behaviour change in limiting Mpox transmission for the reasons given above, it is not possible to establish this beyond reasonable doubt using only the data currently available. At least one recent modelling study is able to explain the case incidence trend in most countries without requiring the effect of behaviour change or public health interventions[21]. To achieve a better understanding the role of different model features we performed a set of sequential forecasts using redacted data at different time points using different model assumptions (see section S.6.2 in Supporting Information for details). We found that although a modelling approach without behaviour change was capable of fitting the whole weekly case incidence data set given complete data, it performed worse than our main model at forecasting incidence trends. A study that linked sexual activity, GBMSM status, Imvanex vaccination status, and Mpox seropositivity would be highly informative in making more concrete conclusions about the relative importance of behaviour change, vaccination uptake and population immunity in driving the Mpox epidemic in the United Kingdom.

The relative simplicity of the model and gaps in the epidemiological data lead to limitations of our work. The delay between symptom onset, seeking treatment, case confirmation and reporting has changed over the course of outbreak; for example, symptom onsets across all of April were included in the first week of May reporting, which could influence our parameter inference. We assume that the probability of detecting infected individuals is constant over time, therefore this model could erroneously attribute the effect of changing case detection rate to behavioural change—although given the severity of some symptoms we do not expect a major change in case detection over time. A shift in detection probability would suggest that a model fitted only on data before the shift would give systematically biased forecasts when compared to holdout data from after the shift. There isn't strong evidence of this effect in the sequential forecasts for the main model (see section S.6.2 in Supporting Information for details). Additionally, we note that the model projections for the peak in cases among non-GBMSM people is early by a couple of weeks, this could be due to our model lacking realistic degrees of social separation that might be captured by a more nuanced network-based transmission model.

The key public health conclusions from our analysis are: that the current case data suggest that Mpox infection is unlikely to be sustained outside the GBMSM population (other transmission $R_0 \sim 0.0378$) although sporadic cases may still occur; that public awareness of Mpox and subsequent behaviour change has had a substantive impact on the Mpox trajectory in the United Kingdom, although we expect this effect to dissipate over time; finally that the vaccine rollout, and the ability to encourage GBMSM people with high sexual contact rates, were important to reduce the risk of Mpox resurgence in the United Kingdom over the medium term. The low level of transmission we infer outside of the GBMSM population is consistent with empirical estimates suggesting low risk of infection to younger, potentially more vulnerable individuals, following exposures in schools[22]. The longer-term dynamics are likely to be governed by the replenishment of susceptible individuals into the highest sexual activity groups, the deployment of vaccine to such individuals and the level of imports from countries with higher incidence.

## Methods

We simulated MPXV transmission in the United Kingdom as a dynamical process where the underlying population at risk was represented as integer sized and subdivided by GBMSM status (GBMSM or non-GBMSM), with the GBMSM population further subdivided by frequency of sexual activity, and into multiple, randomly-sized sub-populations. The daily dynamics of the spread of MPXV were encapsulated in a series of discrete events: transmission, incubation, and recovery, which were assumed to occur stochastically. Each week a sub-sample of individuals that had their symptom onsets the previous week are reported as cases, which connects the underlying transmission model to the observable data.

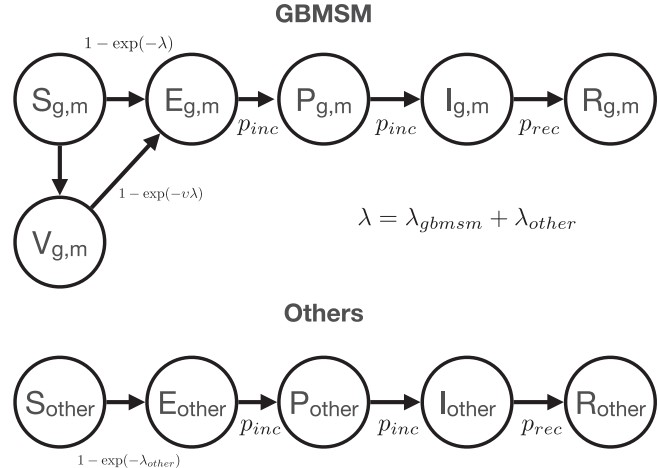

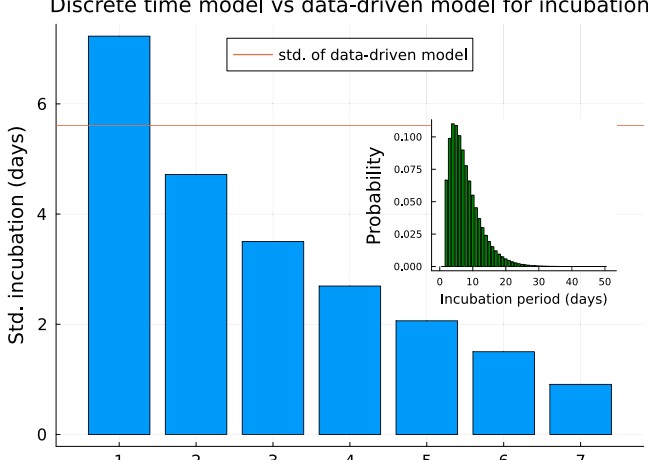

Discrete time model vs data-driven model for incubation

**Fig. 3 | Schematic diagram of compartmental model with daily transition probabilities.** *Top:* Compartmental model for GBMSM individuals. GBMSM population structure is indicated by subscripts for sexual activity group *g* and metapopulation group *m*. Force of infection on GBMSM individuals is sum of infection rate from within GBMSM sexual contacts and background homogeneous transmission (*inset equation*). *Bottom*: Compartmental model for other non-GBMSM individuals. Force of infection on other individuals is only due to background homogeneous transmission.

**Fig. 4 | Multi-stage model for incubation period.** Standard deviation for multi-stage discrete time incubation model when mean is fixed to be 7.7 days by number of stages. The standard deviation for incubation period fitted from data is given as a horizontal red line. The closest match (*n* = 2 stages) was used in simulations. Inset: the incubation period distribution used in simulations.

The design philosophy of this MPXV transmission model was to:
(1) Create a sufficiently parsimonious representation of the MPXV transmission structure that Bayesian posteriors for model parameters could be inferred within reasonable time with limited computational resources.
(2) Capture the important features of heavy-tailed sexual contact networks, by sub-dividing the GBMSM population into sexual activity groups by their rate of forming new sexual partnerships.
(3) Capture the effect of any additional population structure using a random sized metapopulation subdivision of the GBMSM population; metapopulation models being known to be reasonable approximations to more detailed individual based models[19].

The modelling approach in this paper is a hybrid of two well-known transmission model types. If there was only one metapopulation then the transmission model used here would be a random partnership model[23], whereas if there was only one sexual activity group then the transmission model used here would be a metapopulation model[24]. MPXV spread via other pathways than newly formed GBMSM sexual partnerships is modelled as a simple homogeneous transmission process.

## Infectious episode progression model

We model the progression of MPXV infection as a *SEPIR* compartmental epidemic model[23]: susceptible individuals (*S*) contract MPXV and are infected without being infectious (*E*) before becoming pre-symptomatically infectious (*P*) and then symptomatically infectious (*I*), during which periods they can infect other individuals and thus generate further cases. After the actively infectious period individuals recover (*R*) and remain immune to reinfection over the remaining simulation period. The period spent in both *E* and *P* classes defines the incubation period for the model; that is the time between infections and symptoms. A schematic plot for the model is shown as Fig. 3.

Before the 2022 Mpox outbreak the bulk of the MPXV literature assumed that the latency period (duration between infection and becoming actively infectious) and the incubation period (duration between infection and developing symptomatic disease) were the same; that is that infected individuals are infectious when they show symptoms[6]. However, recent studies challenge this view; in this study, we use Ward et al. which find that the incubation period for Mpox,

using posterior mean estimates for parameters, was Weibull (1.4, 8.5) distributed and that the mean serial interval for Mpox was 9.25 days[25].

For a daily probability $p_{inc}$ of progressing between successive stages of our model's *n*-stage incubation period, the number of days spent in the incubation class will be given by $n + d$, where the distribution $f$ of $d$ is negative binomial:

$$f(d|n, p_{inc}) = \binom{d + n - 1}{d} p_{inc}^n (1 - p_{inc})^d \qquad (1)$$

(since $d$ is just the number of days in the incubation period where the individual does not progress to the next stage of infection). Using the method of moments to find a negative binomial distribution with identical mean, and minimised difference in standard deviation, to the Weibull distribution from Ward et al.[25], we found that a two stage incubation process with a daily probability of $p_{inc} = 0.258$ of transitioning between successive stages of infection provided an optimal match between the negative binomial model and the Weibull distribution (average incubation period is $n + \bar{k} \sim 7.7$ days matching the Weibull mean 7.7 days, standard deviation $\sqrt{(1-p_{inc})n/p_{inc}^2} = 4.7$ vs. Weibull deviation 5.6 days, Fig. 4).

The next generation time distribution for the model is the probability distribution of a secondary infection time *W* due to a primary infection; that is the probability $w(d)$ that a secondary infection is generated by a primary infected $d$ days after infection being:

$$w(d) = \frac{\epsilon P(X(d) = P) + P(X(d) = I)}{\sum_{d'} \epsilon P(X(d') = P) + P(X(d') = I)}, \qquad (2)$$

where $X(d)$ is the state of the primary infectee on day $d$ after infection, which in this model is either *E*, *P*, *I*, or *R*, and $\epsilon$ is the relative infectiousness of pre-symptomatic individuals (*P*) compared to symptomatic infectious individuals (*I*). The probability of the primary infected person being in any given state on day $d$ after infection can be calculated by solving the Markov chain associated with the state progression model. While it is likely that infected individuals are infectious whilst symptoms persist, which is typically 2–4 weeks[9], we make two assumptions to operationalise the Mpox transmission model whilst respecting data on incubation period and serial intervals for Mpox:

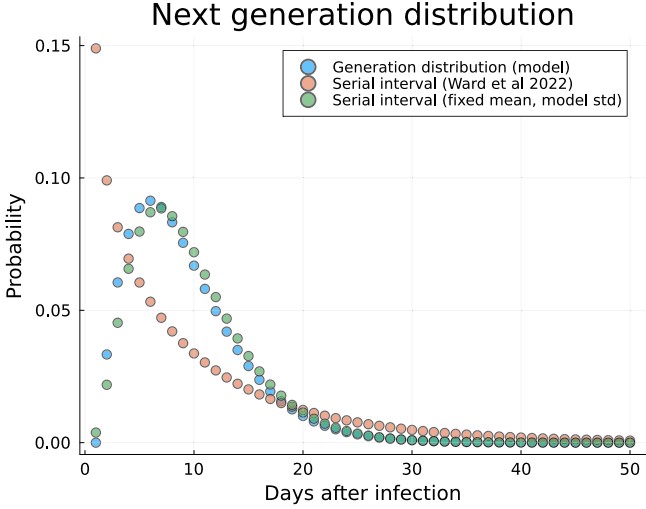

**Fig. 5 | The next generation time distribution used in the model.** Next generation distribution used in this model (*blue dots*) compared against serial interval in Ward et al.[25] (*red dots*) and the same serial interval with reduced standard deviation to match generation distribution used in this model (*green dots*).

- The effective infectious period is self-limiting due to infected individuals reducing their contacts in response to deteriorating MPXV symptoms.
- The expected number of infections generated by a primary infected during their pre-symptomatic phase and the symptomatic phase was the same. The proportion of pre-symptomatic transmission has been hard to estimate (cf Ward et al.[25]), but is potentially substantial[26]. To reduce sensitivity to this parameter we do not explicitly model transmission as being truncated by symptom onset, which would give the pre-symptomatic infectious period a critical role in dynamics, and instead assume that this was fixed by fixing the mean generation period.

Given these assumptions, then $\epsilon$ was determined by solving Eq. (2) such that the mean generation time $\mathbb{E}[W] = 9.5$ days, therefore matching the mean serial interval reported in Ward et al. This gave an estimate of $\epsilon = 0.78$. We define an effective infectious period over both the pre-symptomatic and symptomatic durations as the infectiousness weighted summed durations, $(\epsilon/p_{inc}) + (1/p_{rec}) = 6.01$ days. The next generation time distribution used in the model is shown in Fig. 5.

## Population structure

We subdivided the population of the United Kingdom ($N = 67.2$ million) into gay, bisexual and men who have sex with men (GBMSM) and non-GBMSM. We further subdivided the GBMSM population so that each person belonged to one of 10 sexual activity groups $g = 1, ..., 10$ and one of a random number of metapopulation sub-groups $m = 1, 2, ...$. The non-GBMSM population is not further subdivided.

The proportion of the over 18 year old male population identifying as Lesbian, gay and bisexual (LGB) in the UK has been estimated as 3.4%[27], and the proportion of MSM having at least one new sexual contact with another man in a year, has been estimated as 84.6%[12]. We combine these to make a crude estimate of the size of the sexually active GBMSM population in the UK, $N_{GBMSM} = 760{,}839$.

The population distribution of sexual contacts per year for members of the sexually active GBMSM community, $k$, has previously been estimated as a power law $f(k) \sim k^{-1.81}$[12]. We assumed that the maximum number of sexual contacts per year was 3650, which defined a proper distribution of yearly sexual contacts; if infinitely large $k$ values were allowed then given the power exponent is relatively small the variance would also be infinite, the strict upper bound prevents this from occurring.

In a slight abuse of notation, we assume that each GBMSM individual $i$ has a continuous propensity to form sexual partnerships $k_i$ drawn i.i.d. from $k_i \sim f(k)$. Sexual partnerships form between two GBMSM individuals $i$ and $j$ according to a Poisson process with rate $k_i k_j / (N_{GBMSM} - 1)\langle k \rangle$ per year, therefore, an individual with propensity $k_i$ is expected to have $k_i \sim k^{-1.81}$ distinct sexual partners in a year, respecting the observed power-law distribution in yearly partners. Because the infectious period of Mpox is quite short (in the order of weeks), we don't model the duration of sexual partnerships as in Whittles et al.[12], but rather treat new sexual partnerships as points at which infection can occur as per the random partnership model[23].

We discretized over individual level variation in propensity by dividing the GBMSM population into 10 sexual activity groups by the partitioning the yearly contact rate $1 = k_1 < k_2 < ... < k_9 < k_{10} \leq 3650$ such that:

$$\int_{k_i}^{k_{i+1}} k \times f(k) \mathrm{d}k = E[k]/10, \qquad \forall i \in 1, 2, ..., 10. \qquad (3)$$

By dividing according to Eq. (3) the expected rate of new sexual partnerships was equal across groups, which minimises the loss of information in sexual partnership formation implied by discretising the active GBMSM population. The size of the most active of the sexual activity groups (over the entire metapopulation) was only ~400 out of 760,839 people (~0.05% of the GBMSM population in UK). In general, this defined both $p_g$, the proportion of GBMSM typically in each sexual activity group, and, $\mu_g$, the mean daily rate of creating a new sexual contact conditional on being a member of a sexual activity group $g$ (Fig. 6).

The random sized metapopulation structure was generated according to a Dirichlet multinomial distribution, that is a multinomial where the vector of choice probabilities is drawn from a Dirichlet distribution, $n_{meta} \sim$ DirichletMultinomial($N_{GBMSM}, \alpha_m 1$), where $\alpha_m$ is a real-valued dispersion parameter, 1 is a length 50 vector of ones, and $n_{meta}$ is the resulting vector of metapopulation sizes. Any metapopulation with size 0 was then eliminated from the simulation. It should be noted that $\alpha_m \rightarrow 0$ implies that with probability 1 there will only be one metapopulation of size $N_{GBMSM}$, whereas in the limit $\alpha_m \rightarrow \infty$ the metapopulation size distribution is asymptotically multinomial distributed with on average equal sized metapopulations with mean size $N_{GBMSM}/50 = 15{,}216$.

After generating the randomly sized metapopulations, each metapopulation is further subdivided into sexual activity groups according to a multinomial sample on $p_g$. That is, the generated population size of the sexual activity group $g$ in the metapopulation $m$ ($N_{g,m}$) is conditionally distributed $N_{g,m} | n_{meta} \sim$ Multinomial($n_{meta}[m], p_g$).

A new random metapopulation structure was generated for each simulation, with $\alpha_m$ a target parameter for inference (see subsection *Population structure*).

## Vaccination modelling

UKHSA has secured thousands of vaccines which are being offered to frontline healthcare workers, contacts of cases, and LGB men at highest risk[28]. Within the model we interpret LGB men at highest risk as people within the GBMSM group who typically have a new sexual contact at least once a month (sexual activity groups 3–10, representing the 9.4% of most sexually active MSM people; Fig. 6).

The effectiveness of smallpox vaccine against Mpox has been estimated as 85%[5,29]. We interpret this as the efficacy of smallpox vaccine against acquisition of Mpox rather than just an endpoint efficacy against disease, although it is not possible to distinguish between the two in the data that is publicly available. The proposed dose regime in the UK is to give as many first vaccine doses as possible to LGB men at highest risk, with second doses to be given later as supplies become

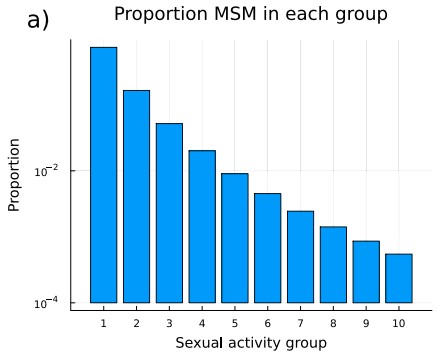

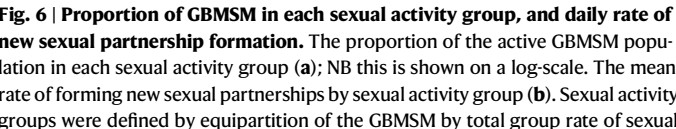

**Fig. 6 | Proportion of GBMSM in each sexual activity group, and daily rate of new sexual partnership formation.** The proportion of the active GBMSM population in each sexual activity group (**a**); NB this is shown on a log-scale. The mean rate of forming new sexual partnerships by sexual activity group (**b**). Sexual activity groups were defined by equipartition of the GBMSM by total group rate of sexual activity (see section *Population structure* for details). The main scenario in this paper assumes that the GBMSM individuals being offered vaccination are those who typically have at least one new sexual partner per month (shown as red line). For this model structure this corresponds to sexual activity groups 3–10 (the 9.4% of the active GBMSM population most sexually active).

available[28]. It is possible that vaccination against Mpox will be less efficacious than 85% against acquisition under this dose regime, therefore whenever modelling the effect of vaccines we draw a vaccine effectiveness parameter $v_{eff} \sim \mathcal{U}(0.7, 0.85)$ at the start of each simulation. We model the action of the vaccine as "leaky"[30]; that is that all vaccinated individuals have their risk of contracting Mpox reduced by $v_{eff}$ per infectious contact (see Eq. (5)).

It has been logistically challenging to capture exact numbers of vaccines that have been taken up by LGB men. However in London (capital city of the United Kingdom) the reported number of vaccines delivered to GBMSM people was around 1000 on the weekend of the 16th/17th July, was expected to be around 2000 on weekend of 23rd/24th July, and sufficient vaccines had been ordered to offer around 5000 doses each weekend in August[18]. In our modelling we assume that the NHS meets these targets in London, and that an additional 67.5% of vaccines are accepted by GBMSM people outside London (Fig. 7), which aligns with the cumulative number of vaccine doses reported as given to GBMSM individuals by 30th August 2022[31]. However, then the uptake rate decreases to 650 doses per week across the country which aligns with the cumulative number of doses reported by 22nd September 2022[32] (Fig. 7). The UK government has not committing to buying more than 100k vaccine doses, therefore, we limit the number of first doses to 50k, that is about 6.5% of the GBMSM population who have typically at least one new sexual partner a year. We do not explicitly model follow-up second doses in this paper which will have a longer term effect on the population immunity.

Within the transmission model the dynamics of vaccine deployment are as follows:

- Within the model vaccination acts to move people from the susceptible (S) group to the vaccinated (V) group (Fig. 3).
- All vaccine doses occur at the end of each week, with the number of vaccines given following the schedule described above.
- Vaccinated individuals are treated as reaching their maximum one dose protection $v_{eff}$ after a 1 week delay.
- We assume that vaccines are only offered prophylactically; that is to individuals without any previous Mpox symptoms (S, E, P epidemiological categories). We assume that vaccines given to cryptically infected people are ineffective.
- The number of vaccines sought by different GBMSM sexual activity groups is proportional to their group size.

**Modelling behavioural change due to Mpox epidemic**
Behavioural response and attitude to risk can significantly affect an epidemic trajectory, as also seen in the COVID-19 pandemic[20,33]. We model behavioural change as leading to lower transmission rate per

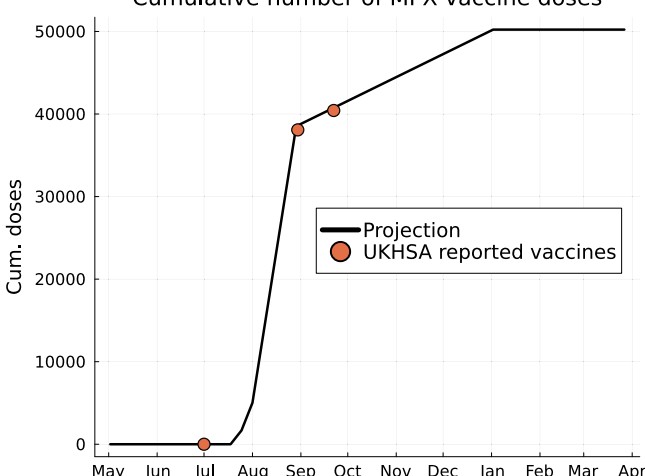

**Fig. 7 | Projected number of cumulative first vaccine doses given in the United Kingdom.** Projected cumulative numbers of first vaccine doses given is shown (*black lines*) compared against UKHSA reported cumulative vaccine first doses (*red dots*).

infected person. Within this model, we assume a fairly rapid lowering of transmission rate over 2 weeks with an unknown midpoint $T_1$, and reversion to pre-outbreak baseline as happening continuously over time with a midpoint for reversion $T_r$ that depends of whether reversion mainly occurs over 4 or 12 weeks (the two scenarios considered in the paper). Additionally, we assumed that there could be a discontinuous change in behaviour at the point when the World Health Organisation announced that Mpox was public health emergency of international concern.

To summarise:

(1) The midpoint of decreased transmission rate on date $T_1$ between 1st May 2022 and 18th July 2022. $T_1$ is a target for inference.

(2) $T_2 = 23$rd July 2022, the date of the announcement by the WHO that Mpox represented a public health emergency of international concern (PHEIC)[2].

(3) $T_r = 15$th September 2022 (4 week reversion), or, 13th October 2022 (12 week reversion).

The effect of behavioural change is assumed to reduce transmission for infectious individuals in both the GBMSM and non-GBMSM populations through some combination of:

- Voluntary self-isolation during MPX symptoms.

- The effect of any treatments against MPX.
- The effect of contact tracing in raising awareness of their exposure, and guiding safer behaviours.
- The social avoidance of other individuals towards those with MPX symptoms.

At the behavioural change points the probability of infection per new sexual contact between GBMSM individuals ($p_{\text{GBMSM}}(t)$) and the reproductive number for other routes of transmission ($R_{\text{other}}(t)$) decrease by some proportion (see subsection "Force of infection"); these proportions of transmission decrease were a target for inference along with the timing of the change point $T_1$.

The probability of transmission per sexual contact $p_{\text{GBMSM}}(t)$ varied over time because of behavioural change (see above) as follows:

$$\frac{p_{\text{GBMSM}}(t)}{p_{\text{GBMSM}}(0)} = \left[1(t \geq T_2)(1 - \rho_{\text{GBMSM},2})\right]\left(1 - \rho_{\text{GBMSM},1}\sigma((t - T_1)/\kappa_1)\right) \\ + \Delta\rho_{\text{GBMSM}}\sigma((t - T_r)/\kappa_r). \tag{4}$$

where $\rho_{\text{GBMSM},1}$ and $\rho_{\text{GBMSM},2}$ are the proportional risk reductions in GBMSM sexual contacts due to behavioural change and a reaction to the WHO announcement of a public health of emergency of international concern, respectively, with $\Delta\rho_{\text{GBMSM}} = 1 - (1 - \rho_{\text{GBMSM},1})(1 - \rho_{\text{GBMSM},2})$ being the proportionate change from 1st May to the minimum point after $T_2$. $\sigma(x) = 1/(1 + e^{-x})$ is the logistic curve governing reversion to baseline risk. $\kappa_1$ was chosen so that transmission rates had decreased by 1% a week before $T_1$ and by 99% a week after $T_1$. Similarly, $\kappa_r$ was chosen so that behaviour reversion to baseline was at 1% above the minimum point on 1st September 2022 and at 99% 2 or 6 weeks after the reversion midpoint $T_r$, depending on whether most reversion was assumed to occur over 4 or 12 weeks.

## Force of infection

We consider two transmission pathways for MPX: (1) transmission during close contact when GBMSM individuals form new sexual partnerships, and (2) all other routes of transmission, including non-GBMSM sexual partnerships and stable GBMSM sexual partnerships, household cohabitation and other known transmission pathways for MPXV. The force of infection for each pathway was, respectively:

$$\hat{I}_{g,m}(t) = \epsilon P_{g,m} + I_{g,m},$$
$$\hat{I}(t) = \epsilon P(t) + I(t),$$
$$\lambda_{\text{GBMSM}}(g,m,t) = \frac{1}{10}\sum_{g',m'}p_{\text{GBMSM}}(t)\mu_{g'}T_{m,m'}\hat{I}_{g',m'}(t)/N_{g,m}, \tag{5}$$
$$\lambda_{\text{other}} = \gamma_{\text{eff}}R_{\text{other}}\hat{I}(t)/N.$$

Here $P_{g,m}(t)$ and $I_{g,m}(t)$ are, respectively, the number of presymptomatic and symptomatic infectious GBMSM people in sexual activity group $g$ and metapopulation $m$, $P(t)$ and $I(t)$ are the total number of presymptomatic and symptomatic infectious people across the whole population (GBMSM and non-GBMSM), and $N$ is the total UK population size. We define the between metapopulation contact structure $T_{m,m'}$ as having 99% of sexual contacts within group, with the rest distributed to other subpopulations by size:

$$T_{m,m'} = 0.99\delta_{m,m'} + 0.01 \times (1 - \delta_{m,m'})\left(N_m / \sum_{m' \neq m}N_{m'}\right). \tag{6}$$

This implicitly sets the meaning of the metapopulation model. The reason for the factor of 10 in the denominator of Eq. (5) is due to, by construction, sexual partnerships being spread equally between the different sexual activity groups.

Note that in the limit $\alpha_m \to 0$, this transmission model recovers the random partnership (or configuration) model with respect to new

sexual formation[23,34]; this is slightly obscured by the construction of the sexual activity groups, that is that each group has the same rate of sexual contacts overall even though the higher activity groups have far fewer members. Therefore, parameter inference is capable of recreating the random partnership model as the most plausible explanation for spread among active GBMSM, as well as allowing for more complicated transmission structure within the GBMSM community if that is a more plausible explanation.

Based on the underlying rates, the daily probability of becoming infected for (1) unvaccinated GBMSM in sexual activity group $g$ and metapopulation $m$, (2) vaccinated GBMSM in sexual activity group $g$ and metapopulation $m$ and (3) non-MSM people, respectively are:

$$P_{inf}(t|gbmsm,g,m,unvac) = 1 - \exp\{-[\lambda_{\text{GBMSM}}(g,m,t) + \lambda_{\text{other}}(t)]\},$$
$$P_{inf}(t|gbmsm,g,m,vac) = 1 - \exp\{-(1 - \upsilon_{\text{eff}})[\lambda_{\text{GBMSM}}(g,m,t) + \lambda_{\text{other}}(t)]\}.$$
$$P_{inf}(t|other) = 1 - \exp\{-\lambda_{\text{other}}(t)\}.$$

It should be noted that we have not included the possibility of external infections in our force of infection, e.g. by including an external forcing term in Eq. (5). Mpox imports into the United Kingdom were likely to have been concentrated early in the epidemic, which we believe is captured by inferring initial conditions (see Table S.3 in Supporting Information).

## Case detection model

We assume that those MPX cases which are detected have a one week reporting lag after onset of symptoms (i.e. around a 7-day delay with 1–14 day delays possible), which is broadly in-line with the estimated reporting delay in July 2022[35]. Cases are differentiated by GBMSM and non-GBMSM but not by underlying metapopulation or sexual activity group. We modelled the number of cases as being a Beta-Binomial distributed sample over the simulated onsets in the previous week. This is probabilistically equivalent to Binomial sampling but with the probability of detection being independently Beta distributed each week. This is a robust approach to Bayesian inference which reduces the effect of outliers on inference[36], and it has been suggested that stochastic components to detection rate can improve inference in epidemiological modelling by absorbing some of the effect of model misspecification[37], which could be important in this model because in April/May 2022 the reporting delay was probably longer[35]. However, because each week's detection probability is drawn from an independent Beta distribution our model will not capture temporal directional trends in case detection, for example a trend towards lower chance of detection over time.

The number of GBMSM and non-GBMSM (other) cases observed in each week $w$ is:

$$C_{\text{GBMSM}}(w) \sim \text{BetaBinomial}\left(\sum_{g,m}O_{\text{GBMSM}}(g,m,w-1),p_d,\phi_d\right)$$
$$C_{\text{other}}(w) \sim \text{BetaBinomial}\left(O_{\text{other}}(w-1),p_d,\phi_d\right)$$

where $O_{\text{GBMSM}}(g,m,w-1)$ was the simulated number of symptom onsets in week $w-1$ in sexual activity group $g$ and metapopulation $m$ of the GBMSM population, $O_{\text{other}}(w-1)$ was the simulated number of symptom onsets in week $w-1$ in the non-GBMSM population, $p_d$ was the mean value of the weekly Beta distributed detection rate, and $\phi_d$ is a dispersion parameter for the weekly Beta distributed detection rate. Given $n$ onsets in a week the mean and variance in number of cases in the next week are, respectively, $np_d$ and $np_d(1 - p_d)(1 + (n - 1)d)$. The more common Beta($\alpha, \beta$) parameterisation can be recovered via the relationships, $p_d = \alpha/(\alpha + \beta)$ and $\phi_d = 1/(\alpha + \beta + 1)$.

## Data and parameter inference

Data on confirmed Mpox cases in UK is maintained by UKHSA and include patients' characteristics, such as their basic demographics (age

and sex), clinical and laboratory records, contact data, and travel histories and gender (GBMSM or non-GBMSM) obtained from questionnaires - although this full spectrum of information is not available for all cases.

Weeks $w = 1, 2, 3, \ldots$ were labelled by their Monday date, and all confirmed cases reported were aggregated by week. For each week $w$, we considered the numbers $C^*_{\text{GBMSM}}(w)$ and $C^*_{\text{other}}(w)$ of reported cases that identify as GBMSM or non-GBMSM, respectively, and the number $C_{\text{NA}}(w)$ of cases for which GBMSM information is missing. Missing values were handled with an imputation method based on gradient boosted decision trees (GBDTs). GBDTs are machine learning models consisting of an ensemble of single decision trees, each including a series of nodes representing binary decision splits against one of the predictor variables[38,39]. GBDTs were trained to learn the probability that a case identifies as GBMSM, given all other available data (Supplementary Information section S1.7).

The probabilities of GBMSM for the cases to be imputed were also averaged by week, in order to estimate the weekly fraction $p(w)$ of GBMSM cases in $C_{\text{NA}}(w)$ (Fig. A8) as follows:

$$
\begin{aligned}
C_{\text{GBMSM}}(w) &= C^*_{\text{GBMSM}}(w) + p(w) \times C_{\text{NA}}(w), \\
C_{\text{other}}(w) &= C^*_{\text{other}}(w) + (1 - p(w)) \times C_{\text{NA}}(w).
\end{aligned}
\tag{7}
$$

While this implied that $C_{\text{GBMSM}}(w)$ and $C_{\text{other}}(w)$ were not necessarily integer valued, the error measure we used in our inference (see Eq. (8)) did not require integer reference data.

We performed Bayesian inference on the model parameters (see Table S.3 in Supporting Information for full list of parameters, priors used and posterior mean and 95% CIs) using sequential Monte Carlo based approximate Bayesian computation (SMC-ABC[40]) implemented in the Julia language package ApproxBayes.jl[41].

Forward simulations were performed by solving the stochastic Mpox transmission model using the DifferentialEquations.jl ecosystem of dynamical system solvers for the Julia programming language[42]. After drawing model parameters from the prior distributions, simulations were initialised at the beginning of the week immediately previous to the week with first reported cases (Monday 25th April 2022, $w = 0$) as follows:

(1)  A random metapopulation and sexual activity group distribution for GBMSM people was generated (see subsection "Population structure").

(2)  One metapopulation was randomly selected proportionally to metapopulation size.

(3)  For the selected metapopulation a Poisson distributed number of individuals were assigned to each incubation stage and the infectious stage (uniformly likely) in each sexual activity group (proportional to population frequency) such that conditional on the chosen $p_d$ and inf parameters the expected number of GBMSM cases on week $w = 1$ was $\iota_0$, which was a target parameter for inference.

During each simulation a predicted (integer) number of reported cases for GBMSM and non-GBMSM on each week was generated using Eq. (5): $\hat{C}_{\text{GBMSM}}(w), \hat{C}_{\text{other}}(w)$ for $w = 1, 2, 3, \ldots$. The error metric for the simulation used by the SMC-ABC algorithm against the true data was $d_1(\hat{C}, C)$, defined as:

$$
d_1(\hat{C}, C) = \frac{\sum_w |\hat{C}_{\text{GBMSM}}(w) - C_{\text{GBMSM}}(w)|_1 + \sum_w |\hat{C}_{\text{other}}(w) - C_{\text{other}}(w)|_1}{\sum_{w'} C_{\text{GBMSM}}(w') + C_{\text{other}}(w')}
\tag{8}
$$

where $|\cdot|_1$ denotes L1 norm. The last week of available case data was not used in inference because of potentially confounding right-censoring.

We used mainly uninformative or flat priors for parameters. The exceptions were: (1) $R_{\text{other}} \sim \text{LogNormal}(\log(0.25), 1)$ which reflected our a priori belief that $R_{\text{other}}$ was likely to be substantially less than one in the United Kingdom setting, because there were no previous examples of large outbreaks despite Mpox being a persistent problem in sub-Saharan countries with frequent transit to-from the United Kingdom, (2) $M = \alpha + \beta \sim \text{Gamma}(3, 1000/3)$ for the Beta-Binomial link (Eq. (5)) to observations which reflected our belief that the case data from the UKHSA would have low overdispersion (we originally formulated the observation model using noisier open data sources), and (3) $\iota_0 \sim \text{LogNormal}(\log(5), 1)$ which reflected our prior belief that the original numbers of infected was in the 10s of people rather than the 100s of people.

Before running SMC-ABC we performed prior predictive model checking and simulation-based calibration for the error target value (see S.2 in Supporting Information).

### Reporting summary
Further information on research design is available in the Nature Portfolio Reporting Summary linked to this article.

## Data availability
This study was conducted for the purpose of informing the UKHSA's outbreak response to the monkeypox pandemic. Work was undertaken using de-anonymised data in line with national data regulations. Raw UKHSA linelist data is not immediately available, however, the post-processed weekly case time-series with GBMSM status imputation (see section S.7 in Supporting Information), along with analysis scripts are available at the public github repository: https://github.com/SamuelBrand1/MpoxUK[43]. The full linelist is not immediately available because UKHSA has a duty to protect sensitive case data. UKHSA operates a robust governance process for applying to access protected data that considers: The benefits and risks of how the data will be used. Compliance with policy, regulatory and ethical obligations. Data minimisation. How the confidentiality, integrity, and availability will be maintained. Retention, archival, and disposal requirements. Best practice for protecting data, including the application of 'privacy by design and by default', emerging privacy conserving technologies and contractual controls. Access to protected data is always strictly controlled using legally binding data sharing contracts. UKHSA welcomes data applications from organisations looking to use protected data for public health purposes and will consider applications on a reasonable timeframe. To request an application pack or discuss a request for UKHSA data you would like to submit, contact DataAccess@ukhsa.gov.uk.

## Code availability
All code used in running the model, performing inference, and creating visualisation is available at the public github repository: https://github.com/SamuelBrand1/MpoxUK[43].

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

## Acknowledgements

This work has benefited from conversations with the UK Health Security Agency (UKHSA) DEA Cell, MPX modelling cell and technical committee. We would like to thank UKHSA colleagues Dan Todkill, Luke Hounsome, and Leo Loman. S.P.C.B., J.H., L.M.G.-R., M.J.K. and D.J.N.'s work was supported by funding from UK Foreign, Commonwealth and Development Office (FCDO) and Wellcome Trust (grant # 220985/Z/20/Z). M.J.K. and T.H. were supported by the UKRI through the JUNIPER modelling consortium (grant no. MR/V038613/1). T.H. was also supported by the Engineering and Physical Sciences COVID-19 scheme (grant number EP/V027468/1), the Royal Society (grant number INF/R2/180067), and the Alan Turing Institute for Data Science and Artificial Intelligence. M.J.K. and M.C.'s work was supported by Health Data Research UK, which is funded by the UK Medical Research Council, EPSRC, Economic and Social Research Council, Department of Health and Social Care (England), Chief Scientist Office of the Scottish Government Health and Social Care Directorates, Health and Social Care Research and Development Division (Welsh Government), Public Health Agency (Northern Ireland), British Heart Foundation and the Wellcome Trust. M.J.K. is also funded by the National Institute for Health Research (NIHR) [Policy Research Programme, Mathematical and Economic Modelling for Vaccination and Immunisation Evaluation, and Emergency Response; NIHR200411]. M.J.K. is affiliated to the National Institute for Health Research Health Protection Research Unit (NIHR HPRU) in Gastrointestinal Infections at University of Liverpool in partnership with UK Health Security Agency (UKHSA), in collaboration with University of Warwick. M.J.K. is also affiliated to the National Institute for Health Research Health Protection Research Unit (NIHR HPRU) in Genomics and Enabling Data at University of Warwick in partnership with UK Health Security Agency (UKHSA). The views expressed are those of the author(s) and not necessarily those of the NHS, the NIHR, the Department of Health and Social Care or UK Health Security Agency, or any other body that funds this work.

## Author contributions

Model concept design: S.P.C.B., M.C., J.H., L.M.G.-R., D.J.N., and M.J.K. Model code design: S.P.C.B. and M.C. Data imputation design and coding: M.C. UKHSA linelist data processing: M.C., T.H., F.C., C.T., I.F., and P.B. Manuscript writing: S.P.C.B., M.C., F.C., J.H., L.M.G.-R., T.H., D.J.N., and M.J.K. Results visualisation design: S.P.C.B., M.C., D.J.N., and M.J.K.

## Competing interests

The authors declare no competing interests.
