## [Peer Review File · Nature Communications]

The role of vaccination and public awareness in forecasts of Mpox incidence in the United KingdomEditorial Note: This manuscript has been previously reviewed at another journal that is not operating a transparent peer review scheme. The manuscript was considered suitable for publication without further review at *Nature Communications*.

REVIEWERS' COMMENTS

Reviewer #1 (Remarks to the Author):

Thank you. The authors have addressed all my comments.

Reviewer #3 (Remarks to the Author):

Thank you for addressing the comments on the previous version of this paper; I am glad that some of the comments proved helpful. I appreciate the substantial effort that has been put into further analyses and clarifications, and find the manuscript stronger as a result.

In particular, I find the additional sensitivity analyses to be helpful for understanding the role and relative importance of the different mechanisms / structural features modelled. The revised §4.3 is also much clearer about how vaccination is implemented in the model.

A few further thoughts:

1) The primary public health insight remains that vaccination is important for preventing a rebound in cases as behaviour reverts to pre-outbreak norms. That said, the importance of prioritizing high-risk individuals for vaccine allocation is also a valuable public health message. While not exactly surprising, the demonstration / quantification of the importance of prioritizing (vs. uniform allocation) is quite stark and convincing.

As such I believe this result deserves greater prominence. For instance, in §3, paragraph 4, there is only a brief mention of 'inefficiently targeted vaccines' – this would be a good place to elaborate on the importance of appropriate vaccine targeting. Similarly, the last paragraph of §3 would be a good place to further emphasize this point.

2) Regarding the 'no behaviour change' model having a better forecast than other models for the late projection date (§A6.2 / Fig. A5) – the data appear to show a slowdown in the decline in weekly cases around September. Is it possible that this was due to some degree of behavioural reversion in reality, which (I presume) is excluded from the forecasts of the other models?

It may also be worth examining the best-fit parameters for the 'no behaviour change' model in that case to see if they are realistic and reasonable. In my experience, when the data show a single neat epidemic peak, model estimation processes will sometimes [erroneously] fit a model without some behaviour change or similar mechanism to reduce transmission rates by estimating unrealistic parameter values or population sizes, thereby fitting the data via depletion of susceptible population. I don't think this is necessarily happening with your model, but it is worth a check.

3) Three minor corrections:

A) §3, paragraph 3, first sentence: "Our model-based analysis it that [sic] the most likely reason..."

B) The table references in §A3 are incorrect

C) Per the comment on language in the previous review, I suggest changing the figure legend labels in Fig A5/A6 to avoid the contraction 'homo. pop.'

Thanks again for this valuable work!

Tse Yang Lim, Ph.D.

Center for Communicable Disease Dynamics
Harvard T.H. Chan School of Public Health

Response to reviewers of “The role of vaccination and public awareness in forecasts of Mpox incidence in the United Kingdom”

Reviewer #1 (Remarks to the Author):

Thank you. The authors have addressed all my comments.

Response: Thank you for reviewing our work, your comments have really improved this study.

Reviewer #3 (Remarks to the Author):

Thank you for addressing the comments on the previous version of this paper; I am glad that some of the comments proved helpful. I appreciate the substantial effort that has been put into further analyses and clarifications, and find the manuscript stronger as a result.

In particular, I find the additional sensitivity analyses to be helpful for understanding the role and relative importance of the different mechanisms / structural features modelled. The revised §4.3 is also much clearer about how vaccination is implemented in the model.

Response: Thank you for reviewing our work, your comments have substantially improved this study.

A few further thoughts:

1) The primary public health insight remains that vaccination is important for preventing a rebound in cases as behaviour reverts to pre-outbreak norms. That said, the importance of prioritizing high-risk individuals for vaccine allocation is also a valuable public health message. While not exactly surprising, the demonstration / quantification of the importance of prioritizing (vs. uniform allocation) is quite stark and convincing. As such I believe this result deserves greater prominence. For instance, in §3, paragraph 4, there is only a brief mention of ‘inefficiently targeted vaccines’ – this would be a good place to elaborate on the importance of appropriate vaccine targeting. Similarly, the last paragraph of §3 would be a good place to further emphasize this point.

Response: We agree that this finding slipped out of focus in the manuscript. We have added “*In particular, we find that failing to target vaccination towards the most at-risk groups would have been almost as ineffective as a no vaccination counter-factual scenario.*” To paragraph 4 of Discussion (section 3) and modified the final paragraph in Discussion to say “...; finally, that the vaccine rollout, and the ability to encourage GBMSM people with high sexual contact rates, were important to reduce the risk of Mpox resurgence in the United Kingdom over the medium term.”.

2) Regarding the ‘no behaviour change’ model having a better forecast than other models for the late projection date (§A6.2 / Fig. A5) – the data appear to show a slowdown in the decline in weekly cases around September. Is it possible that this was due to some degree of behavioural reversion in reality, which (I presume) is excluded from the forecasts of the other models?

Response: This is possible; however, other explanations are possible. For example, a subtle variation in reporting delay at the epidemic peak causing a mild over-estimation on the behaviour change magnitude. We don’t believe that this kind of problem causes systemic problems for the model because, as we describe in section 3, this would be revealed by biases in all the sequential forecasts.

However, we don’t think that it would add value to the main model to modify the structure further.

It may also be worth examining the best-fit parameters for the ‘no behaviour change’ model in that case to see if they are realistic and reasonable. In my experience, when the data show a single neat epidemic peak, model estimation processes will sometimes [erroneously] fit a model without some behaviour change or similar mechanism to reduce transmission rates by estimating unrealistic parameter values or population sizes, thereby fitting the data via depletion of susceptible population. I don’t think this is necessarily happening with your model, but it is worth a check.

Response: This is a good suggestion. Whilst the Bayesian estimation we performed does exclude completely unrealistic parameter values via choosing prior distributions for the parameters, nonetheless there is a notable difference in the inferred case detection probability between the main model structure presented in the paper and the “no behaviour change” model variant considered in the supporting information.

The explanation for the epidemic curve in the “no behaviour change” model is based on only population immunity, with a posterior expected value for the typical chance of case detection of 21.2% (12.1 – 49.9% CI), whereas inference using the main model found a posterior case detection probability of 45.4% (16.7- 86.9% CI). Therefore, the “no behaviour change” model predicts roughly double the number of underlying infections compared to the main model whilst fitting to the same epidemic curve, and hence explains the curve due to extra unobserved population immunity compared to the main model which includes reduction in transmission rates.

Whilst the “no behaviour change” model estimate for case detection rate is further from our prior expectation of case detection, our prior beliefs on case detection were weakly held (we used $p_d \sim \text{Beta}(5, 5)$ as our prior for this parameter in Bayesian inference). We don't believe we can strongly dismiss the “no behaviour change” model based on unrealistic parameter inference.

Nonetheless, we prefer the main model presented in the paper based on coherence with studies of behaviour change among GBMSM people cited in the paper, but not used in inference, and the better forecast accuracy of the main model for most of the epidemic period. These arguments are made more substantially in section A.6.3.

3) Three minor corrections:

- A) §3, paragraph 3, first sentence: “Our model-based analysis it that [sic] the most likely reason...”
- B) The table references in §A3 are incorrect
- C) Per the comment on language in the previous review, I suggest changing the figure legend labels in Fig A5/A6 to avoid the contraction ‘homo. pop.’

Response:

- A) We have corrected this to “Our model-based analysis suggests that...”
- B) This is corrected.
- C) We have changed the legend from ‘homo. pop.’ to ‘one metapopulation’.

Thanks again for this valuable work!

Tse Yang Lim, Ph.D.
Center for Communicable Disease Dynamics
Harvard T.H. Chan School of Public Health